# Association of aflatoxin B1 levels with mean CD4 cell count and uptake of ART among HIV infected patients: A prospective study

**Pauline E. Jolly**[1‡]*, **Tomi F. Akinyemiju**[2©], **Swati Sakhuja**[1©], **Roshni Sheth**[1]

**1** Department of Epidemiology, School of Public Health, University of Alabama at Birmingham, Birmingham, Alabama, United States of America, **2** Department of Population Health Sciences, School of Medicine, Duke University, Durham, North Carolina, United States of America

© These authors contributed equally to this work.
‡ This author is the senior author on this work.
* jollyp@uab.edu

## Abstract

### Background

Aflatoxin suppresses cellular immunity and accentuates HIV-associated changes in T- cell phenotypes and B- cells.

### Objective

This prospective study was conducted to examine the association of aflatoxin levels with CD4 T-cell count and antiretroviral therapy uptake over time.

### Methods

Sociodemographic and food data were collected from antiretroviral therapy naïve HIV-infected patients. CD4+ counts were collected from participants' medical records. Plasma samples were tested for aflatoxin $B_1$ albumin adducts, hepatitis B surface antigen, and HIV viral load. Participants were separated into high and low aflatoxin groups based on the median aflatoxin $B_1$ albumin adduct level of 10.4 pg/ml for data analysis.

### Results

Participants with high aflatoxin $B_1$ albumin adduct levels had lower mean CD4 at baseline and at each follow-up period. Adjusted multivariable logistic regression analysis showed that higher baseline aflatoxin $B_1$ adduct levels were associated with statistically significant lower CD4 counts (est = -66.5, p = 0.043). Not starting ART and low/middle socioeconomic status were associated with higher CD4 counts (est = 152.2, p<0.001) and (est = 86.3, p = 0.027), respectively.

### Conclusion

Consistent correlations of higher aflatoxin $B_1$ adduct levels with lower CD4 over time indicate that there is an independent early and prolonged effect of aflatoxin on CD4 even with

**Data Availability Statement:** All relevant data are within the manuscript and its Supporting information files.

**Funding:** This research was supported by USAID grant LAG-G-00-96-90013-00 for the Peanut Collaborative Research Support Program, University of Georgia and the Minority Health International Research Training Grant #5 T37 MD 001448 from the National Institute on Minority Health and Health Disparities, National Institutes of Health, Bethesda, MD, USA.

**Competing interests:** All authors declare that they have no competing interests.

the initiation of antiretroviral therapy. The prospective study design, evaluation of baseline and follow-up measures, extensive control for potential confounders, and utilization of objective measures of aflatoxin exposure and CD4 count provide compelling evidence for a strong epidemiologic association that deserves careful attention in HIV care and treatment programs.

## Introduction

Aflatoxins are carcinogenic metabolites produced in food crops primarily by two species of *Aspergillus* fungi, namely *A. flavus* and *A. parasiticus* [1]. These fungi are ubiquitous in soil and on vegetation and produce toxins in a variety of staple food crops such as cereals (e.g. maize, millet, rice, and wheat), legumes and oilseeds (soybean and groundnuts), and a number of other crops such as tree nuts, root and tuber crops and spices [2]. In tropical and subtropical regions of the world, the high temperatures and high humidity favor aflatoxin contamination of crops. In addition, poor post-harvest practices and storage and marketing conditions in many countries of sub-Saharan Africa, Latin America, and South and Southeastern Asia result in fungal proliferation and accumulation of aflatoxin levels in crops that exceed the 20 μg/kg limit for total aflatoxins in foods set by the United States [3]. Consequently, outbreaks of acute aflatoxicosis have occurred with the death rate as high as 39.4% [4–6].

Of the four main aflatoxin chemotypes (AFB$_1$, AFG$_1$, AFB$_2$, AFG$_2$), AFB$_1$ is generally the predominant and most toxic form [7]. AFM$_1$ is a major harmful metabolite of AFB$_1$ that is excreted in urine and milk [8, 9]. Although chronic exposure to aflatoxin is known predominantly for its role in the development of liver cancer in humans, especially in those with hepatitis B and hepatitis C infections [10–13], several animal and human studies show that aflatoxins modulate the immune system, mainly causing immune suppression [14–16]. Studies conducted in humans with chronic exposure to dietary aflatoxin show that blood levels of aflatoxin B$_1$ albumin adducts (AF-ALB) are associated with antibody and cellular immunity [17–19].

Human immunodeficiency virus (HIV) infection suppresses the immune system and results in the development of acquired immunodeficiency syndrome (AIDS) unless those infected are treated with antiretroviral therapy (ART). Investigation of immune status of HIV-positive people chronically exposed to aflatoxin in their diets, showed significantly lower percentages of CD4+ T regulatory cells, naïve CD4+ T-cells, perforin-expressing CD8+ T-cells, and B-cells in those with high AF-ALB compared to those with lower AF-ALB levels [19]. These findings indicate that changes in T-cell phenotypes and B-cells that occur in HIV are amplified by aflatoxin exposure. In cross-sectional studies, we also found consistent strong associations between high AF-ALB levels and high HIV viral loads, indicating that aflatoxin exposure may contribute to high viral loads and faster progression to AIDS [20, 21]. Therefore, we conducted this prospective study among ART-naïve HIV-positive asymptomatic Ghanaians with mean baseline CD4 counts of 631±281 cells/mm$^3$ of blood to examine the association of AF-ALB levels in blood with changes in CD4 cell count and uptake of ART over a five-year period.

## Methods

### Study site, participants and data collection

This study was conducted in the Kumasi South Regional Hospital (KSRH) and Bomso Hospital (BH) in Kumasi in the Ashanti Region of Ghana. KSRH is a large hospital that serves 56 communities of approximately 400,000 people. BH is a 163 bed private hospital located in

close proximity to KSRH that provides comprehensive HIV care, treatment, and support and works closely with KSRH. Potential study participants were HIV-positive patients with CD4 count ≥300 cells /mm$^3$ blood who were asymptomatic had received no antiretroviral therapy in accordance with the World Health Organization 2006 treatment guideline [22]. Clinic staff told patients attending the hospitals of the study and asked if they would be interested in participating. Patients who expressed interest in participating were introduced to the study team who explained the study and asked them to read the informed consent and ask questions. After all patient questions were answered to their satisfaction those willing to participate gave written informed consent. An interviewer-administered questionnaire that was developed for the study has been included as S1 File. It contained questions related to sociodemographic, health, food acquisition, storage and consumption practices, awareness of aflatoxin and information on HIV/AIDS and sexually transmitted diseases. Eight staff members (doctors, nurses, and administrative personnel) from both hospitals reviewed the questionnaire for understanding, clarity and cultural appropriateness after which it was revised. It was then pilot tested among six clinic patients similar to the ones recruited for the study and revised before use. To ensure confidentiality, the interviews were conducted in private rooms at the hospitals. Data on HIV diagnosis date and CD4+ T cell count were collected from medical records of the patients. The mean, standard deviation (SD) and range of CD4 cells at recruitment were mean ± SD = 618.10 ± 284.32; range 301–1616. A 20 mL blood sample was collected from each patient and plasma was prepared and tested for AF-ALB, HIV viral load, and HBV surface antigen (HBsAg). CD4 and ART initiation data were collected for up for five years post-recruitment.

## Ethical approval

Ethical approval for the study was obtained from the Institutional Review Board at the University of Alabama at Birmingham (UAB) and the Committee on Human Research, Publications and Ethics at the School of Medical Sciences, Kwame Nkrumah University of Science and Technology (KNUST), Kumasi, Ghana. Written informed consent was obtained from each participant.

## Quantification of AFB$_1$-lysine adducts

Aflatoxin B$_1$-lysine adducts in plasma of participants was measured using a modified High Performance Liquid Chromatography (HPLC)-fluorescence method developed by Qian, et al. (2013) and outlined in detailed in Jolly, et al. (2015) [23, 24]. Aflatoxin B$_1$-lysine adducts indicate exposure to aflatoxin in the previous 2–3 months [25].

## Quantitation of HIV viral load using the Roche HIV-1 RNA Assay

HIV-1 RNA in the plasma of study participants was measured at the UAB Hospital Laboratory using the Roche COBAS Ampliprep/COBAS TaqMan HIV-1 Test, version 2.0 according to the manufacturer instructions (Roche Molecular Systems, Inc, Pleasanton, CA). The test has been approved by the United States Food and Drug Administration, and quantifies HIV-1 RNA based on the co-amplification of the HIV LTR (Long Terminal Repeat) and *gag*. This method has been outlined in detail in Jolly et al. (2013) [21].

## Test for antibodies to HBV surface antigen

Antibodies to HBsAg in plasma samples were determined using the Bio-Rad Enzyme Immunoassay according to the manufacturer's directions (Bio-Rad, Redmont, WA, USA) and previously outlined by Jolly et al. (2011) [20].

## Statistical analyses

Two hundred and ninety-five participants who had complete data on baseline AF-ALB levels are included in the current analysis. Participants were divided into high and low AF-ALB levels based on the median AF-ALB level of 10.4pg/mg. We describe sociodemographic characteristics, food consumption patterns, and clinical variables of participants by high ($\geq$10.4 pg/mg) and low (<10.4 pg/mg) AF-ALB levels at baseline using chi-square tests for categorical variables, fisher's exact test for cell counts <5, and t-tests to compare group means for continuous variables. Principal components analysis (PCA) was conducted to assess lasting household indicators such as housing type, plumbing, water, and electricity to determine the socioeconomic status (SES). An SES score was attained from the PCA analysis by weighting each indicator by the coefficient of the first principal component, with each member of the household being assigned the same SES. Further, SES was categorized into tertiles ranging from lowest to highest. Mixed methods analysis was employed in Statistical Analysis System (SAS; SAS Institute Inc., Cary, North Carolina, USA) to analyze this longitudinal data to assess statistically significant predictors of CD4 levels over time from baseline to 5 years. Initial adjustments included sociodemographic variables (age, gender, and SES score), season (dry and rainy), baseline AF-ALB levels and knowledge of HIV-positive status. A second model was additionally adjusted for alcohol intake and food consumption patterns. A final model included further adjustment for health status, viral load, HBV status, and ART. For all analyses, p-values $\leq$0.05 were considered as statistically significant. All statistical analyses were performed with SAS 9.4.

## Results

The mean AF-ALB level for the participants in this study was 14.75 pg/mg (standard deviation ±15.61); median = 10.37, range = 0.20–109.87, and inter-quartile range = 4.67–19.56 pg/mg. The median AF-ALB was used to separate participants into high and low AF-ALB groups (Table 1). A majority of participants were 30–39 years of age and married/cohabitating. A significantly higher proportion of males had high baseline AF-ALB levels as compared to females (73.0% vs. 44.0%; p<0.001). Significantly higher mean viral load (176,264 vs. 73,900; p = 0.010) was observed among participants with high AF-ALB as compared to those with low AF-ALB (Table 1). Among participants with high AF-ALB, 61.6% reported buying 20% or more of their food as compared to 49.6% of participants with lower AF-ALB (p = 0.045). A higher proportion of participants with high AF-ALB (44.2%) reported storing 25% to >50% of maize as compared to 25.9% of participants with low AF-ALB (p = 0.005). A higher percent of participants with high AF-ALB also reported storing maize for 3 to >6 months (41.0% vs. 23.2%; p = 0.006).

Table 2 shows the unadjusted mean CD4 estimates during baseline, and at 1 to 5 years of follow-up visits by sociodemographic variables and season. A lower mean CD4 count was observed among those with high levels of baseline AF-ALB and at each collection period as compared to those with low AF-ALB levels, with the lowest levels during the 5th year (533.1 ±41.5 vs. 755.0±38.8). Fig 1 shows the mean CD4 estimates over time by baseline AF-ALB levels. Participants who started ART during the study period had a lower baseline mean CD4 count as compared to those who did not start ART (483.9±24.8 vs. 702.4±21.7); although, by year 5 of follow-up, the mean CD4 count was higher among those on ART as compared to those who did not start ART (666.6±34.5 vs. 526.5±41.3). Additionally, males and those of high SES were observed to have lower mean CD4 counts. Older participants ($\geq$40 years) had lower mean CD4 counts at baseline and during years 1–3. Lower mean CD4 counts were also observed during the dry seasons at baseline and years 1–4 as compared to rainy seasons

**Table 1. Descriptive statistics of the study participants by median baseline aflatoxin albumin adduct (AF-ALB) levels.**

| | High AF-ALB ($> = 10.4$pg/mg) N = 147 | Low AF-ALB ($<10.4$pg/mg) N = 148 | P value |
|---|---|---|---|
| **Age** | | | 0.972 |
| 18–29 | 41 (27.9) | 43 (29.0) | |
| 30–39 | 67 (45.6) | 67 (45.3) | |
| 40 and above | 39 (26.5) | 38 (25.7) | |
| **Gender** | | | **<0.001** |
| Female | 102 (69.4) | 132 (89.2) | |
| Male | 45 (30.6) | 16 (10.8) | |
| **Marital status** (missing = 6) | | | 0.735 |
| Married/ Cohabitating | 100 (69.9) | 100 (68.5) | |
| Separated/Divorced/Widowed | 23 (16.1) | 21 (14.4) | |
| Single | 20 (14.0) | 25 (17.1) | |
| **Socioeconomic status** (missing = 13) | | | 0.380 |
| Low | 48 (33.8) | 40 (28.6) | |
| Middle | 52 (36.6) | 48 (34.3) | |
| High | 42 (29.6) | 52 (37.1) | |
| **Religion** (missing = 6) | | | 0.981 |
| Christian | 124 (86.1) | 125 (86.2) | |
| Muslim/Other | 20 (13.9) | 20 (13.8) | |
| **Proportion of food grown** (missing = 21) | | | 0.243 |
| None | 77 (56.2) | 85 (62.0) | |
| <20% | 28 (20.4) | 31 (22.6) | |
| $> = 20\%$ | 32 (23.4) | 21 (15.4) | |
| **Proportion of food bought** (missing = 18) | | | **0.045** |
| <20% | 53 (38.4) | 70 (50.4) | |
| $> = 20\%$ | 85 (61.6) | 69 (49.6) | |
| **Proportion of maize stored** (missing = 18) | | | **0.005** |
| <25% | 77 (55.8) | 103 (74.1) | |
| 25%-49% | 55 (39.9) | 31 (22.3) | |
| $> = 50\%$ | 6 (4.3) | 5 (3.6) | |
| **Months maize stored** (missing = 18) | | | **0.006** |
| 0–2 | 82 (59.0) | 106 (76.8) | |
| 3–5 | 52 (37.4) | 29 (21.0) | |
| $> = 6$ | 5 (3.6) | 3 (2.2) | |
| **Groundnut consumption** (missing = 20) | | | 0.278 |
| Never | 9 (6.5) | 7 (5.1) | |
| Once or less a week | 63 (45.3) | 68 (49.6) | |
| 2–3 times a week | 48 (34.5) | 35 (25.5) | |
| Everyday | 19 (13.7) | 27 (19.7) | |
| **Maize consumption** (missing = 21) | | | 0.050 |
| Never | 1 (0.7) | 2 (1.5) | |
| Once or less a week | 36 (25.9) | 51 (37.5) | |
| 2–3 times a week | 46 (33.1) | 27 (19.9) | |
| Everyday | 56 (40.3) | 56 (41.1) | |
| **Drink alcohol** (missing = 2) | | | 0.487 |
| No | 134 (91.8) | 138 (93.9) | |
| Yes | 12 (8.2) | 9 (6.1) | |
| **Health status** (missing = 5) | | | 0.052 |

(*Continued*)

**Table 1.** (Continued)

| | High AF-ALB (> = 10.4pg/mg) N = 147 | Low AF-ALB (<10.4pg/mg) N = 148 | P value |
|---|---|---|---|
| Poor | 4 (2.8) | 0 (0.0) | |
| Average | 58 (40.3) | 72 (49.3) | |
| Good | 82 (56.9) | 74 (50.7) | |
| **Hepatitis B virus status** (missing = 8) | | | 0.345 |
| Negative | 128 (87.1) | 134 (90.5) | |
| Positive | 19 (12.9) | 14 (9.5) | |
| **HIV viral load**# (missing = 2) | 176264 (453675) | 73900 (129910) | **0.010** |
| **CD4 T cell count**# | 613.6 (281.6) | 644.7 (277.4) | 0.340 |
| **On antiretroviral therapy (ART)** | | | 0.147 |
| No | 105 (71.4) | 94 (63.5) | |
| Yes | 42 (28.6) | 54 (36.5) | |
| **Season** | | | 0.865 |
| Dry | 79 (53.7) | 81 (54.7) | |
| Rainy | 68 (46.3) | 67 (45.3) | |

N = 306 with Missing baseline aflatoxin levels for 11

*SES rank includes: education, employment, housing type, plumbing, water, electricity, house material

#Mean and SD

**Table 2. Unadjusted mean estimates for CD4 count over the study period.**

| | Baseline | 1 Year | 2 Years | 3 Years | 4 Years | 5 Years |
|---|---|---|---|---|---|---|
| | Mean±Std Err | Mean±Std Err | Mean±Std Err | Mean±Std Err | Mean±Std Err | Mean±Std Err |
| **AF-ALB pg/mg (baseline)** | | | | | | |
| Low | 648.0±25.1 | 618.4±34.8 | 610.7±32.1 | 680.8±30.2 | 741.0±30.0 | 755.0±38.9 |
| High | 608.5±23.5 | 611.9±33.1 | 554.7±30.9 | 626.0±29.1 | 629.1±33.9 | 533.1±41.5 |
| **On ART** | | | | | | |
| No | 702.4±21.7 | 695.8±31.8 | 668.3±30.4 | 704.5±28.3 | 627.3±34.7 | 526.5±41.3 |
| Yes | 483.9±24.8 | 480.5±32.0 | 468.8±28.2 | 553.5±26.7 | 677.4±30.4 | 666.6±34.5 |
| **Sex** | | | | | | |
| Female | 651.2±19.0 | 631.4±27.2 | 594.5±25.1 | 654.8±23.6 | 693.4±27.2 | 651.3±31.5 |
| Male | 548.9±39.8 | 538.1±52.3 | 569.5±49.5 | 620.2±46.7 | 631.7±59.3 | 579.8±67.4 |
| **Age (years)** | | | | | | |
| 18–29 | 703.3±32.9 | 677.0±49.9 | 610.1±49.8 | 644.1±44.2 | 637.0±52.6 | 581.7±57.2 |
| 30–39 | 628.4±26.1 | 622.7±35.8 | 630.3±33.7 | 660.2±31.9 | 691.0±38.3 | 629.1±47.8 |
| ≥40 | 552.4±31.1 | 528.1±41.8 | 511.3±36.4 | 629.8±35.3 | 699.0±39.3 | 684.6±43.2 |
| **Socioeconomic status** | | | | | | |
| Low | 638.0±33.3 | 640.4±47.0 | 595.2±43.7 | 709.4±41.8 | 711.8±47.4 | 716.0±56.1 |
| Middle | 654.0±30.6 | 664.3±42.6 | 642.4±40.9 | 662.6±38.6 | 675.4±43.2 | 682.7±50.0 |
| High | 609.2±27.5 | 561.7±37.1 | 552.1±34.0 | 608.5±31.3 | 639.5±39.7 | 549.5±44.1 |
| **Season** | | | | | | |
| Dry | 584.8±20.8 | 575.6±29.9 | 546.2±28.4 | 611.7±25.7 | 632.5±29.6 | 676.1±36.8 |
| Rainy | 683.4±27.9 | 657.6±38.3 | 637.1±34.9 | 692.7±33.9 | 741.9±40.8 | 615.8±43.6 |

ART = antiretroviral therapy

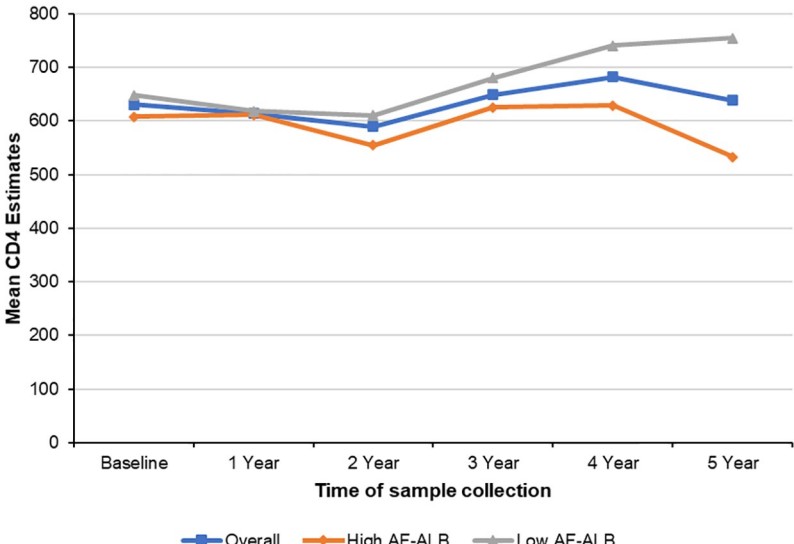

**Fig 1. Mean CD4 estimates over time by baseline AF-ALB levels.**

(Table 2). Fig 2 shows the mean CD4 estimates over time by baseline AF-ALB levels and ART initiation. Fig 3a and 3b show the CD4 distribution with time of ART initiation for the low and high AF-ALB groups, respectively. At baseline, the mean CD4 level was high for the study group and no participant was on ART. At year one the mean CD4 had dropped and 39 participants (15 in the high and 24 in the low AF-ALB groups) started ART, while at year two the mean CD4 was at its lowest and an additional 33 participants (12 in the high and 21 in the low AF-ALB groups) started ART. At year three, the mean CD4 was high but about 16 new

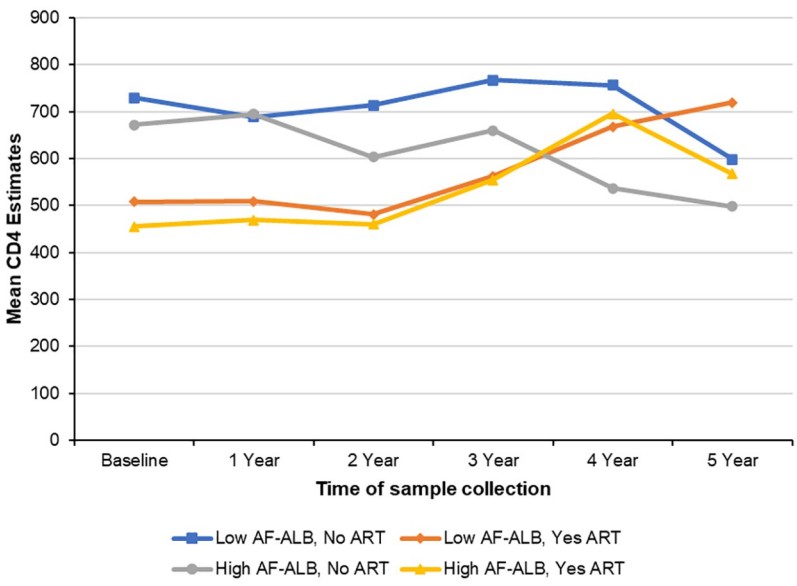

**Fig 2. Mean CD4 estimates over time by baseline AF-ALB levels and ART status.**

(a)

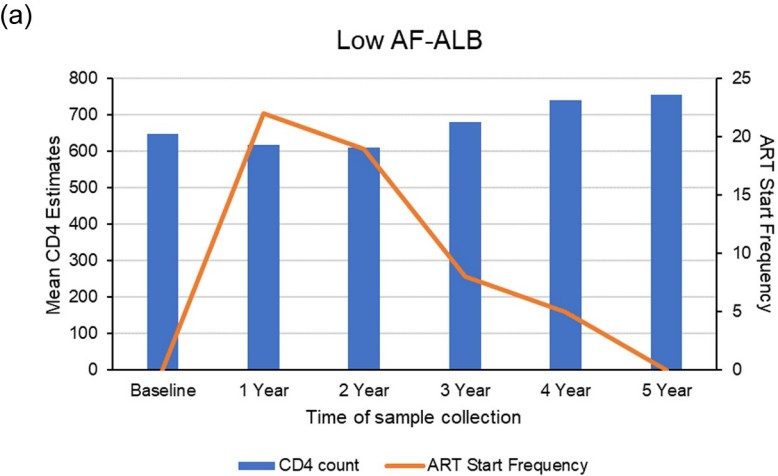

(b)

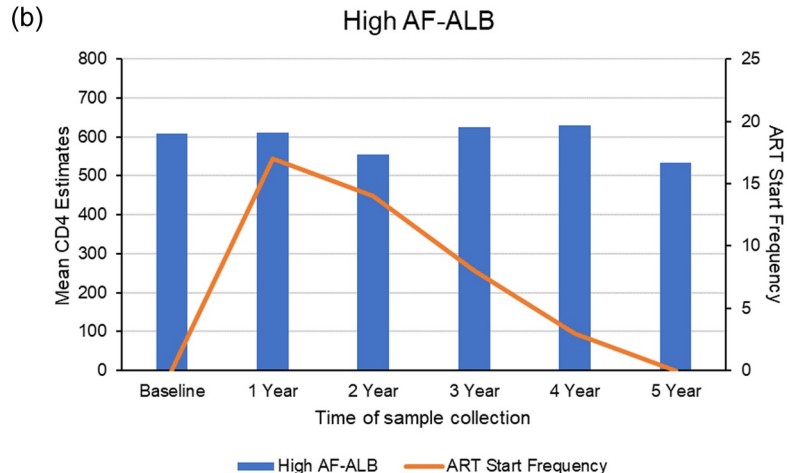

**Fig 3. (a) CD4 count distribution with start time for ART for the low AF-ALB group and (b) CD4 Count Distribution with Start Time for ARTfor the high AF-ALB group.**

participants (3 in the high and 13 in the low AF-ALB groups) started ART. At year four the mean CD4 was at its highest for the group with 8 new participants (3 in the high and 5 in the low AF-ALB groups) starting ART, and at year five no new participant started ART. At year 5, the mean CD4 was much higher than the baseline level for the low AF-ALB group but lower than baseline for the high AF-ALB group.

Table 3 presents the results of the adjusted multivariable models. Model 1 shows that higher baseline AF-ALB was associated with statistically significant lower CD4 counts (est = -66.5, p = 0.043); not starting ART was associated with higher CD4 count (est = 152.2, p<0.001). The ART results remained significant after adjustment for other confounders and clinical variables in Models 2 and 3. Low and middle SES was associated with higher CD4 counts in all models. When adjusted multivariable logistic regression models were run with AF-ALB as a continuous variable, in addition to the results reported above female gender was associated with higher CD4 count.

**Table 3. Multivariable adjusted models of mean CD4 estimates over the study period.**

| | Model 1 | | Model 2 | | Model 3 | |
|---|---|---|---|---|---|---|
| | Beta estimates | P value | Beta estimates | P value | Beta estimates | P value |
| **AF-ALB pg/mg (baseline; high and low based on median of 10.4 pg/mg)** | | **0.043** | | 0.117 | | 0.085 |
| High | -66.5 | | -57.4 | | -61.4 | |
| Low | Ref | | Ref | | Ref | |
| **On ART** | | **<0.001** | | **<0.001** | | **<0.001** |
| No | 152.2 | | 130.9 | | 125.2 | |
| Yes | Ref | | Ref | | Ref | |
| **Gender** | | 0.090 | | 0.053 | | 0.060 |
| Female | 71.2 | | 92.8 | | 93.0 | |
| Male | Ref | | Ref | | Ref | |
| **Age** | | 0.349 | | 0.878 | | 0.812 |
| 18–29 | 17.6 | | -26.4 | | -26.8 | |
| 30–39 | 6.9 | | -31.9 | | -29.7 | |
| 40 and above | Ref | | Ref | | Ref | |
| **Socioeconomic status** | | **0.027** | | **0.028** | | **0.014** |
| Low | 86.3 | | 79.9 | | 95.1 | |
| Middle | 63.3 | | 23.7 | | 36.1 | |
| High | Ref | | Ref | | Ref | |
| **Season** | | 0.137 | | 0.248 | | 0.260 |
| Dry | -45.2 | | -31.8 | | -31.2 | |
| Rainy | Ref | | Ref | | Ref | |

[1]Adjusted for age, gender, SES rank, art status, season, aflatoxin level at baseline, knowledge of HIV-positive status

[2]Adjust for model 1 variables + alcohol consumption + food consumption patterns

[3]Adjust for model 2 variables + health status + viral load + Hepatitis B status

ART = antiretroviral therapy

## Discussion

This study was conducted in a major metropolitan area of Kumasi in the Ashanti Region of Ghana where people are at high risk for exposure to aflatoxin in food. Maize and groundnuts are staple crops in Kumasi, with maize being the principal crop and these crops having the highest aflatoxin contamination [26–28]. Most of the food that participants purchased was grown in rural areas and traded in urban markets. The purchased maize may vary in quality and level of aflatoxin contamination depending on the time reaped and the post-harvest processing and storage methods prior to selling [29]. Additional storage over a longer period under hot and humid conditions by study participants would likely result in fungal proliferation, buildup of aflatoxin levels, and greater intake in food [30].

AF-ALB adduct levels in our study participants ranged from 0.20–109.87 pg/mg. AF-ALB is a reliable biomarker that has been used as a standard for assessment of population exposures. It represents accumulation of adducts from repeated (chronic) exposure to aflatoxin over a 2–3 month period and has been used in epidemiological and clinical intervention studies on aflatoxin exposure in humans in different countries. Studies conducted in regions of the world at high risk of aflatoxin exposure show that more than 95% of people are positive for AF-ALB adducts with concentrations ranging from 3–5 pg/mg albumin to >1000 pg/mg [31].

The most significant finding from this prospective study is that adjusted multivariable logistic regression analysis showed that higher baseline AF-ALB was associated with significantly

lower CD4 counts. The unadjusted data showed that higher AF-ALB was associated with lower mean CD4 at baseline and at each follow-up time point over the 5-year period, with the lowest mean level during the fifth year. This indicates that the effect of aflatoxin occurs early in HIV infection and remains consistent over time even with the initiation of ART. In the third and fourth years of the study as more participants initiated ART, the mean CD4 levels increased but the increases were less among those with high AF-ALB compared to those with low AF-ALB and the difference was much greater at year 5.

Participants reporting higher SES had lower mean CD4 counts at all time-points. This was a significant finding in all of the multivariable logistic regression models. One possible explanation could be that these higher SES participants were diagnosed with HIV after a longer period of HIV infection and/or took charge of their own health care for a longer period before attending the public clinic for HIV care. We observed that in the early to mid-2000s, HIV infected people with better economic means would attend private clinics/hospitals to avoid disclosure of their HIV-positive status. However, there was an agreement at the United Nations General Assembly High-Level Meeting on AIDS in 2006, to scale up HIV prevention, treatment, care, and support services. In response, the Ghana AIDS Commission developed the National Strategic Framework outlining targets to increase ART coverage to 60% and increase the number of persons receiving HIV care by 200% by 2013 [32]. This resulted in the establishment of programs for the provision of ART in public hospitals and health centers in districts in all ten regions of Ghana and removal of the availability of ART from private clinics/hospitals. HIV infected people of higher SES may have taken a longer time to attend the public facilities for HIV care. ART was provided free of cost to all HIV-positive patients in this study who met the WHO recommended guideline for ART in 2009 and updated in 2010 [22].

When adjusted multivariable logistic regression models were run with AF-ALB as a continuous variable, male gender was found to be significantly associated with lower CD4 count. Lower CD4 counts could be due to a longer time of HIV infection before diagnosis among males who access healthcare facilities less frequently than females [33]. However, the models were adjusted for knowledge of when participants knew of their HIV-positive status. Interestingly, a significantly higher percent of males also had high AF-ALB levels at baseline and at each of the 5-year follow-up time points when compared to females, and in previous studies, males were found to have higher AF-ALB levels than females [24, 34]. Animal studies have shown greater effects of mycotoxins on feed intake and weight gain and in clinical and immunological parameters among males [35–37]. It is possible that there is greater susceptibility of males to aflatoxin related to differences in metabolism of mycotoxins by the liver as suggested by previous authors [38, 39] and this may be an indication of the greater immunological effect of aflatoxin among HIV infected males.

It is understandable that participants who initiated ART during the study had significantly lower baseline mean CD4 counts than those who did not initiate ART and that by year 5 of follow-up, the mean CD4 count was higher among those on ART when compared to those who did not initiate ART. Although ART is available in most countries of the developing world where people are chronically exposed to aflatoxin, the number of people who need ART globally has increased as the WHO guidelines change and have become more difficult to meet. In 2018, 62% of HIV-positive people globally and 64% in the WHO Africa Region who needed ART were able to receive it [40]. Therefore, more than one-third of people who need ART are not receiving it. In addition, the pool of people who will need ART will continue to increase as new HIV infections continue to occur; 1.7 million new infections occurred in 2018, of which 800,000 occurred in sub-Saharan Africa) [41]. Furthermore, although deaths from HIV-related illnesses have decreased from the peak of 1.7 million in 2004, 770,000 people died of AIDS-related illnesses mostly in low- and middle-income countries in 2018 [41]. Since our studies

indicate that aflatoxin contributes to an increase in HIV viral load, changes in immune status, and CD4 decline, the full impact of ART is not being, and will not be attained in countries and among those with chronic high exposure to aflatoxin.

Further, HIV-positive people in these high aflatoxin exposure countries who are not diagnosed early, as well as those who do not receive ART, will die faster. It has been shown that HIV-positive people in high aflatoxin-exposure countries progress to AIDS and die more rapidly than those living in high income countries with strict regulation of aflatoxin levels [42]. The median time from HIV seroconversion to clinical AIDS was 11 years for HIV-positive people from Europe, North America, and Australia compared to only 7.4 years in a group of Thai soldiers [42]. A West African study showed that the mean time from HIV seroconversion to clinical AIDS ranged from 5 to 7.2 years depending on the HIV-1 subtype [43]. Since the effect of aflatoxin on increasing HIV viral load and decreasing CD4 is more pronounced in men, and men are more likely to be diagnosed with HIV at a later stage of infection than women, the adverse effects of HIV and disease progression among men can be expected to be greater in aflatoxin exposed areas.

The WHO has set global targets for the elimination of HIV as a public health threat by 2030 [44]. However, the decline in the number of people newly infected with HIV is too slow to reach the goal of 500,000 new infections set for 2020. Combination prevention targets such as increase in voluntary male medical circumcision, condom use, pre-exposure prophylaxis, and harm-reduction services continue to be insufficient in stemming the tide of new infections and financial resources available for the AIDS response have declined. While greater efforts are needed to tackle difficult aspects of HIV that prevent its elimination as a major public health problem, aflatoxin contamination of crops, and dietary exposure to the toxin is a specific problem associated with HIV infection and progression that is overlooked. Several agricultural, dietary and clinical strategies to reduce aflatoxin exposure and its adverse effects on health have been proposed [45]. The cost-effectiveness of these methods has also been elaborated in detail; however, these methods have not been adopted to any extent in countries most affected by aflatoxin exposure [46]. Policy makers, health officials, and researchers should apply this information if they are to be successful in current efforts in combating HIV and other major health problems.

## Limitations

This study has certain limitations that should be considered in interpreting the results. First, a convenience sampling method was used that is prone to inherent bias in representation. As such, the sample may not be representative of the study population and the results may not be generalizable to the population being studied. However, convenience sampling enabled us to study the population in a relatively expedient way and to obtain novel results that would not have been possible otherwise. Secondly, we were unable to evaluate aflatoxin levels in food consumed by study participants, examine their nutritional status, or assess their exposure to other mycotoxins or environmental toxins. In addition, the survey data on food grown, maize storage, and food consumption is likely to have recall bias. However, there are several strengths of this research as well. Foremost, the prospective design allowed us to observe changes in CD4 levels over time and to examine CD4 levels by sociodemographic (age, sex, and SES) and seasonal factors that could not be done using a cross-sectional design. Additionally, the use of the AF-ALB biomarker is a direct and valid measure of aflatoxin exposure in participants over the previous two to three months. Finally, the relatively homogenous dietary pattern among study participants allows for greater generalization of our findings to others in the population.

This study advances previous work by revealing significant consistent correlations between higher AF-ALB and lower CD4 counts every year over a 5-year period, in a unique cohort of ART-naïve HIV infected asymptomatic individuals, from across all SES. This strongly indicates that the effect of aflatoxin occurs early in HIV infection and remains consistent over time even with the initiation of ART. Our prospective study design, evaluation of baseline and follow-up measures, extensive control for potential confounders, and utilization of objective measures of AF-ALB and CD4 counts provide compelling evidence for a strong epidemiologic association that deserves careful attention. Guidelines and assessments to minimize chronic exposure to aflatoxin among people living with HIV should be incorporated into HIV care and treatment programs for optimum effect of ART and the healthiest survival of those affected.

## Conclusions

The finding of the association of significantly higher AF-ALB levels with lower CD4 counts at baseline and over time is a distinctive contribution to the literature on the effect of aflatoxin on CD4 levels in HIV-infected people. The results indicate that aflatoxin has an immunological effect that contributes to decrease in CD4 and that the effect of aflatoxin occurs early in HIV infection and remains consistent over time even with the initiation of ART. Although the mean CD4 levels increased when patients initiated ART, after year four the CD4 dropped among those with high AF-ALB compared to those with low AF-ALB.

## Supporting information

**S1 Table. Multivariable adjusted models of mean CD4 estimates over the study period (with aflatoxin at baseline as continuous variable).**
(DOCX)

**S1 File. Baseline questionnaire aflatoxin and health status in HIV disease.**
(PDF)

**S1 Dataset. Study dataset.**
(XLSX)

## Acknowledgments

We thank the participants and the clinic staff who facilitated the study.

## Author Contributions

**Conceptualization:** Pauline E. Jolly.

**Data curation:** Pauline E. Jolly, Swati Sakhuja, Roshni Sheth.

**Formal analysis:** Pauline E. Jolly, Tomi F. Akinyemiju, Swati Sakhuja, Roshni Sheth.

**Investigation:** Roshni Sheth.

**Methodology:** Tomi F. Akinyemiju.

**Project administration:** Pauline E. Jolly.

**Supervision:** Pauline E. Jolly.

**Visualization:** Pauline E. Jolly.

**Writing – original draft:** Pauline E. Jolly, Tomi F. Akinyemiju, Swati Sakhuja.

**Writing – review & editing:** Pauline E. Jolly, Swati Sakhuja.

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
