## [Decision Letter · Decision Letter 0]

6 May 2021

PONE-D-21-00637

Association of Aflatoxin B1 Levels with Mean CD4 Cell Count and Uptake of ART among HIV Infected Patients: A Prospective Study

PLOS ONE

Dear Dr. Jolly,

Thank you for submitting your manuscript to PLOS ONE. After careful consideration, we feel that it has merit but does not fully meet PLOS ONE’s publication criteria as it currently stands. Therefore, we invite you to submit a revised version of the manuscript that addresses the points raised during the review process. In particular, patient's description and statistical analysis as well as a number or other points that may be of importance.

We look forward to receiving your revised manuscript.

Kind regards,

Isabelle Chemin, PhD

Academic Editor

PLOS ONE

Journal Requirements:

4. Thank you for submitting the above manuscript to PLOS ONE. During our internal evaluation of the manuscript, we found significant text overlap between your submission and the following previously published works.

- https://moam.info/mycotoxins_5a3c12051723dd42662aa122.html

- https://doi.org/10.3390/toxins7124868

- https://doi.org/10.2217/fmb.13.166

We would like to make you aware that copying extracts from previous publications, especially outside the methods section, word-for-word is unacceptable, even for works which you authored. In addition, the reproduction of text from published reports has implications for the copyright that may apply to the publications.

Please revise the manuscript to rephrase the duplicated text, cite your sources, and provide details as to how the current manuscript advances on previous work. Please note that further consideration is dependent on the submission of a manuscript that addresses these concerns about the overlap in text with published work.

Reviewers' comments:

Reviewer's Responses to Questions

**Comments to the Author**

1. Is the manuscript technically sound, and do the data support the conclusions?

Reviewer #1: Partly

2. Has the statistical analysis been performed appropriately and rigorously? 

Reviewer #1: No

3. Have the authors made all data underlying the findings in their manuscript fully available?

Reviewer #1: No

4. Is the manuscript presented in an intelligible fashion and written in standard English?

Reviewer #1: Yes

5. Review Comments to the Author

Reviewer #1: This clinical prospective study was among ART naïve HIV+ patients in Ghana over 5 years between 2009-2013 with relatively high baseline CD4 count with mean in the 600s (wide range of 300-1616). Participants were grouped into two groups, high AF-ALB vs. low AF-ALB levels. The authors observed that higher AF-ALB group at each time point was associated with lower CD4 count and even with multivariable analysis, this association was significant. Data on diet and sociodemographics were based on questionnaires and HIV disease and diagnosis (including CD4 and ART initiation) were based on medical record. Blood was collected for measurement of AF-ALB level (quantifies level in the past 2-3 months) through HPLC, as well as HIV-1 RNA viral load.

The study placed participants into high (63%) and low (37%) AF-ALB groups. The results showed that higher level AF-ALB group, there was a greater proportion of males compared to females; higher mean VL and lower CD4 count observed compared to lower group. Higher level also was correlated with storing maize for longer period of time, buying food.

Overall, the hypothesis and data supporting it is intriguing; reporting a significant observation that higher AF-ALB group associated with lower CD4 count at baseline and continued through year 1-4. If validated, this could be clinically important in guiding food consumption/storage guidelines to minimize exposure to high levels of AF-ALB exposure among PWH.

The study had several important observations:

• The cohort was unique, in that all were treatment naïve and across socieoconomic status with good spread in age groups. There was some uneven distribution in low/middle SES group between the two groups, with more in the high AF-ALB group (67% vs. 60%) which may effect the endpoint measured, like CD4, access to ART and baseline VL.

• Lower CD4 count significantly observed in those with baseline high AF-ALB levels which persisted at each collection period gives a consistent correlation between the two factors.

• Interesting finding of lower CD4 count observed at baseline and throughout years during dry season, which previous study had showed higher AF-ALB levels in dry season. This is intriguing, but could be confounded by many factors not accounted for in this study.

Minor criticisms:

• Survey data about food grown, maize storage and food consumption likely has significant recall bias and unclear if these questionnaire had not been previously validated to be accurate measures.

• More lower + middle SES in the high AF-ALB group may bias results although comparison of SES between groups was not statistically different based on authors analysis. However, if the group was divided into low+middle vs. high there is likely difference between the two groups based on AF-ALB levels. For example, baseline VL was 200K in high vs 83K in low AF-ALB group. This could be biased by SES and related assess to care and ART which is suggestive of greater proportion of high socioeconomic status in the low AF-ALB group (28 vs. 34%). Similarly, lower SES is related to high AF-ALB exposure, but low SES likely means less access to ART or care and so allows, therefore, augmenting the CD4 difference.

• Data needs to be presented on proportion in each group on ART vs. not on ART at each year of follow up as this ART frequency in each group is unclear and could certainly be biasing the CD4 observations. Figure 3 shows ART frequency but not broken down by frequency in each AF-ALB group, which is the basis of all the comparisons.

• Figure 3: Unclear why CD4 dropping in this cohort at year 5 when each year more people were started on ART, this should continue to go up, thus suggesting groups of patients are not being started on ART and therefore will dropping CD4 counts.

• Discussion mentioned many participants were females in “antenatal clinics”; female hormone and pregnancy is known to affected CD4 and VL count and therefore may again bias data. It would be important to know how many of participants were pregnant and recent post-natal because of effects of pregnancy hormones on CD4 count.

Major criticism:

Overall, the study reports a significant association between high AF-ALB level and lower CD4 baseline and recovery. It is purely a correlative study and does not establish causality even though it has the samples and cohort to do so. A major issue with the study is the heterogeneity of the two groups in terms of HIV status (nadir CD4 count/ years of infection, baseline CD4 count, ART status and years on ART while in study) and the unclear rationale for dividing into two groups based on cutoff of 15pg/mg of AF-ALB. I am concern of the strengths of the associations found in this study as these are all important confounders which could have biased results to show an effect when in fact, biologically they may not be.

For example, conclusion that males had lower CD4 count at baseline and then at each follow up timepoint. If they started at lower T cell, they will continue to be lower regardless of AF-ALB level. Although this difference can be due to AF-ALB effect, this observation could simply be due to the fact that males in this cohort were infected for longer periods. Again, showing that duration of HIV infection of each participant (or marker such as nadir CD4 count) would need to be considered in the analysis. Similarly, the observation that age was associated with lower mean CD4 count in the study could again be simply due to difference in duration of HIV infection (which the author acknowledges but if this data was collected then could be adjusted for in analysis). The two groups would have to matched or controlled for differences in baseline CD4 count.

The second major criticism is that the study treated AF-ALB levels as a categorical or dichotomous variable (high vs. low) instead of continuous variable which would have been more informative as I would argue that seems more clinically relevant as we expect in real-life that AF-ALB levels would run a wide range and there will be a concentration effect if there is indeed an immunosuppressive effect on CD4 cells. It is unclear why the 15pg/mg was used as the cut-off to determine the high vs. low categories as the author never described the significance of this chosen level. It would be interesting to run the analysis with AF-ALB levels as a continuous variable and assess for correlation with CD4 count, but need to control duration of HIV infection or nadir CD4 count.

The third major criticism is that the study does not control or separate out analysis of those on ART, many started during study period. They observed that HIV VL was higher and CD4 cell count lower in the high AF-ALB group at baseline but this could be accounted for by the greater proportion of those NOT on ART (69% vs. 66.3%). Looking at CD4 level change over time is important, but not as a group (AF-ALB high vs. low). The difference proportion in each on ART and/or ART initiation in the middle of study is not accounted for and still treated as two groups. It would be more accurate to look at trajectory of CD4 count each group (ie, see if there is difference in rise of CD4 count after 1 years, 2 year, 3 years, etc after ART initiation between AF-ALB high vs. low groups). Making this comparison with a heterogenous and unevenly distributed groups (ART frequency in Figure 3 should be divided into the AF-ALB groups to compare proportion started in each group) is not accurate.

Fourth major criticism of the study is that this is a correlative study and does not establish or even suggest causality. To make a more casual association, authors could assess change in level of AF-ALB in each patient to change in T cell frequency or even test for function (activation/exhaustion markers, etc) over time. This could have been done with this prospective longitudinally followed cohort of 5 years.

6. PLOS authors have the option to publish the peer review history of their article (what does this mean?). If published, this will include your full peer review and any attached files.

Reviewer #1: No

---

## [Author Response · Author response to Decision Letter 0]

15 Jul 2021

Isabelle Chemin, PhD

Academic Editor

PLOS ONE

Re: PONE-D-21-00637

Association of Aflatoxin B1 Levels with Mean CD4 Cell Count and Uptake of ART among HIV Infected Patients: A Prospective Study

PLOS ONE

Dear Dr. Chemin,

Thank you for sending the comments from the reviewers of our paper submitted to PLOS ONE. We have made the corrections requested and have attached the revised paper with highlights for further consideration. We thank the reviewers for their careful review and believe that the changes have significantly improved the quality of the manuscript. This is a point-by-point response detailing the revisions that have been highlighted in the manuscript.

COMMENTS FOR THE AUTHOR:

Response: The laboratory protocols used in this paper have been published previously; the references for the publications are given in the paper. 

Response: We read the PLOS ONE style requirements at the websites above and ensure that our manuscript meets the style requirements.

Response: We have specified that written informed consent was obtained from each participant in the ethics statement in the Methods and in the online submission information. This study did not include minors. 

Response: We have explained that we developed a questionnaire as part of this study. A copy has been included as Supporting Information file S1_File.pdf. 

 4. Thank you for submitting the above manuscript to PLOS ONE. During our internal evaluation of the manuscript, we found significant text overlap between your submission and the following previously published works. 

- https://moam.info/mycotoxins_5a3c12051723dd42662aa122.html

- https://doi.org/10.3390/toxins7124868

- https://doi.org/10.2217/fmb.13.166

We would like to make you aware that copying extracts from previous publications, especially outside the methods section, word-for-word is unacceptable, even for works which you authored. In addition, the reproduction of text from published reports has implications for the copyright that may apply to the publications.

Please revise the manuscript to rephrase the duplicated text, cite your sources, and provide details as to how the current manuscript advances on previous work. Please note that further consideration is dependent on the submission of a manuscript that addresses these concerns about the overlap in text with published work.

Response: We have revised the manuscript to completely rewrite the duplicated text, cite the sources, and provide details as to how the current manuscript advances previous work (pages 18-19). 

Comments to the Author

1. Is the manuscript technically sound, and do the data support the conclusions?

Reviewer #1: Partly 

Response: We have revised the data analysis and the manuscript in response to the reviewer’s critiques and suggestions below to ensure that it is technically sound and that the data support the conclusions.

 2. Has the statistical analysis been performed appropriately and rigorously? 

Reviewer #1: No

 Response: We have re-run and conducted additional statistical analyses in response to the reviewer’s comments below. Tables 1 and 3 have been re-done completely using the median aflatoxin value to divide the study participants into high and low aflatoxin groups. Based on this change, the first two rows on Table 2 have been revised and highlighted. Figure 3 has been changed to into Figures 3a and 3b to show CD4 counts and ART initiation for the low and high AF-ALB groups separately over time. A Supplemental Table has been added for the multivariable logistic models with aflatoxin as a continuous variable. ________________________________________

3. Have the authors made all data underlying the findings in their manuscript fully available?

The PLOS Data policy requires authors to make all data underlying the findings described in their manuscript fully available without restriction, with rare exception (please refer to the Data Availability Statement in the manuscript PDF file). The data should be provided as part of the manuscript or its supporting information or deposited to a public repository. For example, in addition to summary statistics, the data points behind means, medians and variance measures should be available. If there are restrictions on publicly sharing data—e.g. participant privacy or use of data from a third party—those must be specified.

Reviewer #1: No

Response: The data for this study involves human research participant information (including private medical record information of participants) and therefore will be made available upon reasonable request.

4. Is the manuscript presented in an intelligible fashion and written in standard English?

 Reviewer #1: Yes 

Response: We thank the reviewer. 

 5. Review Comments to the Author

Reviewer #1: Reviewer’s description of the study and findings: This clinical prospective study was among ART naïve HIV+ patients in Ghana over 5 years between 2009-2013 with relatively high baseline CD4 count with mean in the 600s (wide range of 300-1616). Participants were grouped into two groups, high AF-ALB vs. low AF-ALB levels. The authors observed that higher AF-ALB group at each time point was associated with lower CD4 count and even with multivariable analysis, this association was significant. Data on diet and sociodemographics were based on questionnaires and HIV disease and diagnosis (including CD4 and ART initiation) were based on medical record. Blood was collected for measurement of AF-ALB level (quantifies level in the past 2-3 months) through HPLC, as well as HIV-1 RNA viral load.

The study placed participants into high (63%) and low (37%) AF-ALB groups. The results showed that higher level AF-ALB group, there was a greater proportion of males compared to females; higher mean VL and lower CD4 count observed compared to lower group. Higher level also was correlated with storing maize for longer period of time, buying food.

Overall, the hypothesis and data supporting it is intriguing; reporting a significant observation that higher AF-ALB group associated with lower CD4 count at baseline and continued through year 1-4. If validated, this could be clinically important in guiding food consumption/storage guidelines to minimize exposure to high levels of AF-ALB exposure among PWH.

The study had several important observations: 

• The cohort was unique, in that all were treatment naïve and across socioeconomic status with good spread in age groups. There was some uneven distribution in low/middle SES group between the two groups, with more in the high AF-ALB group (67% vs. 60%) which may effect the endpoint measured, like CD4, access to ART and baseline VL. 

Response: There was some uneven distribution in AF-ALB in the low/middle vs high SES group (70% vs 63% using the median), therefore our analytical strategy included covariate adjustment for SES to account for potential confounding. We created an SES score using Principal Components Analysis based on household indicators such as housing type, plumbing and electricity, similar to approaches used in studies of LMIC populations. We observed that low/middle SES status was associated with higher CD4 counts. A possible explanation for this is that higher SES participants were diagnosed with HIV after a longer time of HIV infection and/or took charge of their own health care for a longer period before attending the public clinic for HIV care. We observed that in the early to mid-2000s, HIV infected people with better economic means would attend private clinics/hospitals to avoid disclosure of their HIV-positive status. However, in accordance to scale up HIV prevention, treatment, and support services (United Nations General Assembly Meeting on AIDS in 2006), the Ghana AIDS Commission established programs to provide ART in public hospitals and health centers in districts in all ten regions of Ghana and removed of the availability of ART from private clinics/hospitals [34]. HIV infected people of higher SES may have taken a longer time to attend the public facilities for HIV care. Access to ART was not a factor in this study as ART was available free of cost to ALL participants during the time of the study regardless of SES. Baseline viral load was significantly higher in the high AF-ALB group. It is established that high viral load is associated with low CD4. This explanation is included in the discussion on pages 15-16. 

• Lower CD4 count significantly observed in those with baseline high AF-ALB levels which persisted at each collection period gives a consistent correlation between the two factors.

• Interesting finding of lower CD4 count observed at baseline and throughout years during dry season, which previous study had showed higher AF-ALB levels in dry season. This is intriguing, but could be confounded by many factors not accounted for in this study. 

 Response: We have removed the discussion on season from the paper since season was not significant in any of the multivariable models. 

Minor criticisms:

• Survey data about food grown, maize storage and food consumption likely has significant recall bias and unclear if these questionnaire had not been previously validated to be accurate measures.

Response: We agree with the reviewer that there could be recall bias in the survey data on food grown, maize storage and food consumption, and have included this as a limitation in the paper (page 18). The questionnaire was developed for the study and has been included as supporting information. Eight staff members (doctors, nurses, administrative personnel) from the hospitals reviewed it for understanding, clarity and cultural appropriateness after which it was revised. It was then pilot tested among six clinic patients similar to the ones recruited for the study and revised before use in the study.

• More lower + middle SES in the high AF-ALB group may bias results although comparison of SES between groups was not statistically different based on authors analysis. However, if the group was divided into low+middle vs. high there is likely difference between the two groups based on AF-ALB levels. For example, baseline VL was 200K in high vs 83K in low AF-ALB group. This could be biased by SES and related assess to care and ART which is suggestive of greater proportion of high socioeconomic status in the low AF-ALB group (28 vs. 34%). Similarly, lower SES is related to high AF-ALB exposure, but low SES likely means less access to ART or care and so allows, therefore, augmenting the CD4 difference. 

Response: In re-running multivariable model 1 using the median AF-ALB to separate participants into high and low groups, SES is significant. However, contrary to the reviewer’s comments that the findings “could be biased by SES and related access to care and ART which is suggestive of greater proportion of high SES status in the low AF-ALB group (now 30 vs. 37%). Similarly, lower SES is related to high AF-ALB exposure, but low SES likely means less access to ART or care and so allows, therefore, augmenting the CD4 difference”, we found that low/middle SES was associated with higher CD4 counts. We explained on page 4 above in our response to the reviewer that the likely reason for this finding is that HIV infected people of higher SES may have taken a longer time to attend the public hospitals and clinics for HIV care. Access to ART was not a factor in this study as ART was available free of cost to ALL participants during the time of the study regardless of SES. Ghana Government hospitals and clinics provided free HIV care and treatment to all HIV infected patients. 

• Data needs to be presented on proportion in each group on ART vs. not on ART at each year of follow up as this ART frequency in each group is unclear and could certainly be biasing the CD4 observations. Figure 3 shows ART frequency but not broken down by frequency in each AF-ALB group, which is the basis of all the comparisons.

Response: We have added Figures 3a and 3b that present the proportions of participants in the low and high AF-ALB groups on ART vs. not on ART at each year of follow-up. A higher number of participants in the low AF-ALB group went on ART in the first two years. 

• Figure 3: Unclear why CD4 dropping in this cohort at year 5 when each year more people were started on ART, this should continue to go up, thus suggesting groups of patients are not being started on ART and therefore will dropping CD4 counts.

Response: The mean drop in CD4 for the total study group (original Figure 3 submitted with the paper) was 40 cells (from 680 at year 3 to 640 at year 4). The new Figures 3a and 3b show that the drop in CD4 at year 5 occurred among the high AF-ALB group. There could be a number of reasons for this besides access to ART. As a result of their high AF-ALB status, the participants could have developed poorer health over time, developed less ability to tolerate ART regimen, or decreased adherence to ART. ART was available to ALL participants. 

• Discussion mentioned many participants were females in “antenatal clinics”; female hormone and pregnancy is known to affected CD4 and VL count and therefore may again bias data. It would be important to know how many of participants were pregnant and recent post-natal because of effects of pregnancy hormones on CD4 count.

Response: It has been shown that “CD4 counts were an average of 56 cells/mm3 lower during pregnant compared to non-pregnant periods and 70 cells/mm3 lower during pregnant compared to postpartum periods” (Heffron et al. JAIDS 2014; 65(2): 231-236). https://www.ncbi.nlm.nih.gov/pmc/articles/PMC3898601/. This “drop” is only temporary and is not believed to be a real reduction in CD4 cells but the result of the same amount of cells in a larger amount of blood (the amount of blood increases in pregnancy). In our study, we did not collect pregnancy or recent postnatal data for the women. However, our results show that women had higher CD4 counts than men at ALL time points. 

Major criticism:

Overall, the study reports a significant association between high AF-ALB level and lower CD4 baseline and recovery. It is purely a correlative study and does not establish causality even though it has the samples and cohort to do so. 

Response: Typical epidemiologic studies outside of clinical trials are unable to establish causality in any context. However, our prospective study design, evaluation of baseline and follow-up measures, extensive control for potential confounders, and utilization of objective measures of AF-ALB and CD4 counts provide compelling evidence for a strong epidemiologic association that deserves careful attention. 

A major issue with the study is the heterogeneity of the two groups in terms of HIV status (nadir CD4 count/ years of infection, baseline CD4 count, ART status and years on ART while in study) and the unclear rationale for dividing into two groups based on cutoff of 15pg/mg of AF-ALB. I am concern of the strengths of the associations found in this study as these are all important confounders which could have biased results to show an effect when in fact, biologically they may not be. 

Response: In the original paper submitted the mean AF-ALB level of 15pg/mg was used to divide the participants into high and low AF-ALB groups. Since AF-ALB is skewed, we went back and ran all of the analyses using the median AF-ALB level of 10.4pg/mg and present entirely new Tables and Figures. 

For example, conclusion that males had lower CD4 count at baseline and then at each follow up timepoint. If they started at lower T cell, they will continue to be lower regardless of AF-ALB level. Although this difference can be due to AF-ALB effect, this observation could simply be due to the fact that males in this cohort were infected for longer periods.

Response: We removed the discussion on gender from the paper since gender is not significant in the adjusted models using the median. However, when the models were run with AF-ALB as a continuous variable, female gender was significantly associated with higher CD4 count in all of the adjusted models. 

Again, showing that duration of HIV infection of each participant (or marker such as nadir CD4 count) would need to be considered in the analysis. 

Response: We included knowledge of HIV positive status in the new analysis. 

Similarly, the observation that age was associated with lower mean CD4 count in the study could again be simply due to difference in duration of HIV infection (which the author acknowledges but if this data was collected then could be adjusted for in analysis). The two groups would have to matched or controlled for differences in baseline CD4 count. 

Response: We agree with the reviewer, we have removed the discussion on age from the paper. The former discussion was based on unadjusted data. Age is not significant in the adjusted models. 

The second major criticism is that the study treated AF-ALB levels as a categorical or dichotomous variable (high vs. low) instead of continuous variable which would have been more informative as I would argue that seems more clinically relevant as we expect in real-life that AF-ALB levels would run a wide range and there will be a concentration effect if there is indeed an immunosuppressive effect on CD4 cells. It is unclear why the 15pg/mg was used as the cut-off to determine the high vs. low categories as the author never described the significance of this chosen level. It would be interesting to run the analysis with AF-ALB levels as a continuous variable and assess for correlation with CD4 count, but need to control duration of HIV infection or nadir CD4 count. 

Response: We thank the reviewer for this critique and based on the skewedness in AF-ALB levels, we used the median level of 10.4 pg/mg (instead of the mean level of 15pg/mg) to separate participants into low and high groups and re-analyzed the data. We also ran the adjusted multivariable models with AF-ALB as a continuous variable and obtained results similar to that obtained using the median AF-ALB. Additionally, we obtained significant results for gender with females having higher CD4 levels than males in all models (Supplemental Table). We controlled for the time when participants knew of their HIV positive status in the new analysis. 

The third major criticism is that the study does not control or separate out analysis of those on ART, many started during study period. They observed that HIV VL was higher and CD4 cell count lower in the high AF-ALB group at baseline but this could be accounted for by the greater proportion of those NOT on ART (69% vs. 66.3%). Looking at CD4 level change over time is important, but not as a group (AF-ALB high vs. low). The difference proportion in each on ART and/or ART initiation in the middle of study is not accounted for and still treated as two groups. It would be more accurate to look at trajectory of CD4 count each group (ie, see if there is difference in rise of CD4 count after 1 years, 2 year, 3 years, etc after ART initiation between AF-ALB high vs. low groups). Making this comparison with a heterogenous and unevenly distributed groups (ART frequency in Figure 3 should be divided into the AF-ALB groups to compare proportion started in each group) is not accurate. 

Response: We have separated CD4 counts and ART initiation for the high and low AF-ALB groups over time and presented the results in Figures 3a and 3b as the reviewer suggested. 

Fourth major criticism of the study is that this is a correlative study and does not establish or even suggest causality. To make a more casual association, authors could assess change in level of AF-ALB in each patient to change in T cell frequency or even test for function (activation/exhaustion markers, etc) over time. This could have been done with this prospective longitudinally followed cohort of 5 years. 

Response: Please see response on causality above. Observational studies are not methodologically able to establish causality, however our study design approach enabled rigorous control for potential confounders that taken together indicate strong epidemiologic associations. We thank the reviewer for the suggestion about evaluating AF-ALB in relation to T-cell frequency over time. Our goal with the current analysis and manuscript is to present initial findings that serves as the basis for future studies examining T cell function over time. This area of research is markedly understudied, and our findings will shed more light on the biology of AF-ALB exposure in this population. We will explore funding opportunities to conduct additional analysis of immune cell function as a next step.

Thank you for your kind consideration.

Respectfully,

Pauline Jolly, PhD, MPH

Professor Emerita,

Director, UAB Minority Health International Research Training Program

Recipient, 2014 Ellen Gregg Ingalls/UAB National Alumni Society Award for Lifetime Achievement in Teaching 

2018 Fulbright Specialist Scholar, Institute of Public Health, Ho Chi Minh City, Vietnam

---

## [Editor Report · Decision Letter 1]

19 Nov 2021

Association of Aflatoxin B1 Levels with Mean CD4 Cell Count and Uptake of ART among HIV Infected Patients: A Prospective Study

PONE-D-21-00637R1

Dear Dr. Jolly,

We’re pleased to inform you that your manuscript has been judged scientifically suitable for publication and will be formally accepted for publication once it meets all outstanding technical requirements.

Kind regards,

Isabelle Chemin, PhD

Academic Editor

PLOS ONE

Additional Editor Comments (optional):

The answers provided and changes in the manuscript are sound and improved the quality of the manuscript.
---

## [Editor Report · Acceptance letter]

14 Jan 2022

PONE-D-21-00637R1 

Association of Aflatoxin B1 Levels with Mean CD4 Cell Count and Uptake of ART among HIV Infected Patients: A Prospective Study   

Dear Dr. Jolly:

I'm pleased to inform you that your manuscript has been deemed suitable for publication in PLOS ONE. Congratulations! Your manuscript is now with our production department. 

Kind regards, 

on behalf of

Mrs Isabelle Chemin 

Academic Editor

PLOS ONE